# An Unusual Presentation of Synchronous Breast Cancer and Skin Malignancy in a Patient with Lynch Syndrome: A Case Report and Review of the Literature

**DOI:** 10.3390/biomedicines12061242

**Published:** 2024-06-03

**Authors:** Maiar Elghobashy, Michael Siafakas, Mona Elshafie, Rahul Hejmadi, Naren N. Basu, Abeer M. Shaaban

**Affiliations:** 1The Royal Wolverhampton NHS Trust, Wolverhampton WV10 0QP, UK; 2South Birmingham Screening, Queen Elizabeth Hospital Birmingham, Birmingham B15 2GW, UK; 3Cellular Pathology, Queen Elizabeth Hospital Birmingham, Birmingham B15 2GW, UK; 4Oncoplastic Breast Surgery, Queen Elizabeth Hospital Birmingham, Birmingham B15 2GW, UK; 5Cancer and Genomic Sciences, University of Birmingham, Birmingham B15 2TT, UK

**Keywords:** Lynch syndrome, breast cancer, skin cancer, sebaceous carcinoma

## Abstract

Background: Lynch syndrome is an autosomal dominant condition that leads to an increased risk of many neoplasms. In the United Kingdom, NICE recommends that patients with colorectal and endometrial cancer should be tested for Lynch syndrome. There is conflicting evidence in the literature on the link between breast cancer and Lynch syndrome. Case presentation: A 54-year-old woman presented with a lump in her right breast with a background of locally advanced colorectal cancer and Lynch syndrome due to a MLH1 gene mutation. A core biopsy showed a grade 3, invasive, triple-negative NST carcinoma. The tumour was triple-negative with patchy positivity for CK14 and CK5/6. Simultaneously, a cystic skin lesion in the contralateral breast was noted, which comprised lesional cells with a proliferation of clear cells and bland basaloid cells. The lesion had evidence of sebaceous differentiation with AR, podoplanin and p63 positivity. MSH1 and PMS2 deficiency was found in the breast and skin lesions. Conclusions: In Lynch syndrome, it is vital to be aware of the increased risk of various types of cancer. This case adds to the body of evidence of the spectrum of malignancies that can be encountered in patients with Lynch syndrome.

## 1. Introduction

Lynch syndrome (formerly known as hereditary non-polyposis colorectal cancer/HNPCC) is an inherited autosomal dominant condition causing a significantly increased susceptibility to various types of neoplasms, particularly colorectal cancer. The incidence of Lynch syndrome in the United States has been estimated to be 1 in 279 people and is estimated to be between 1:370 and 1:2000 in the Western population. Most patients, however, will not be aware they are affected [1]. The condition is caused by pathogenic variants in mismatch repair genes (MMRs in order of prevalence: *MLH1*, *MSH2*, *MSH6*, *PMS2*, *EPCAM*) [2,3,4]. It is the most common cause of inherited colorectal cancer but is also associated with an increased risk of other cancers, such as endometrial and ovarian cancers [5].

Due to the strong link between Lynch syndrome and both colorectal and endometrial cancer, the UK National Institute of Health and Care Excellence (NICE) recommends that all patients with colorectal or endometrial cancer should be tested for Lynch syndrome using immunohistochemistry or microsatellite instability (MSI) [6].

Multiple sebaceous neoplasms have been described as part of the Muir–Torre variant of Lynch syndrome. Sebaceous adenomas are the most common neoplasms identified, but sebaceous carcinomas and keratoacanthomas have also been reported [7]. The lesions predominantly occur in the periocular region of the face, and few examples have been described outside their periocular region [8]. The majority of extra-ocular cases occur in the head and neck region [9].

There has been conflicting evidence in the literature regarding the link between Lynch syndrome and breast cancer and whether breast cancer fits into the Lynch syndrome spectrum of cancers.

In this report, we present a rare case of breast cancer and sebaceous carcinoma in the contralateral breast in a patient known to have Lynch syndrome. The histological appearances and differentials are considered, and an up-to-date literature review is presented.

## 2. Clinical Presentation

A 54-year-old female presented in January 2023 symptomatically with a lump in her right breast. Her past medical history included pT4 N0 locally advanced colorectal cancer diagnosed 15 years earlier, for which she was managed with extended right hemicolectomy and sleeve gastrectomy. This was followed up with fluoropyrimidine and oxaliplatin adjuvant chemotherapy. Due to her young age at colorectal cancer presentation, she was genetically tested five years later and found to have a pathogenic variant in the *MLH1* gene indicating Lynch syndrome. At that point, she underwent a risk-reducing (prophylactic) hysterectomy with oophorectomy as per the guidelines, which were tumour-free.

On clinical examination of the breast lump, a benign nodule (P2) was noted above the right nipple. A mammogram of the right breast showed a subtle, focal increase in density in the upper central breast. This was then visualised with tomosynthesis, which revealed an ill-defined mass lesion measuring 20 × 11 mm, superior and posterior to the nipple, which was radiologically malignant (M5) (Figure 1a,b). An ultrasound scan demonstrated an irregular hypoechoic mass measuring 25 × 20mm with malignant features (U5) (Figure 1c). An ultrasound-guided core biopsy was taken, and the biopsy confirmed a grade 3, invasive, triple-negative, no special type (NST) carcinoma. Following a multidisciplinary team discussion, she was offered bilateral (right therapeutic and left risk-reducing) mastectomy with right sentinel node biopsy in view of her history of Lynch syndrome.

Two years prior to the presentation, the patient had also noted a dark spot near the inferior mammary fold of her left (contralateral) breast. A subsequent screening mammography and ultrasound concluded that the lesion represented a benign sebaceous cyst (Figure 1d). A biopsy was not performed due to its typical characteristics of a sebaceous cyst and concern over an inflammatory response in the surrounding tissue following rupture [10]. After discussion with the patient regarding the surgical options, including breast-conserving surgery followed by radiotherapy or mastectomy, the patient opted for a bilateral mastectomy, with her history of Lynch syndrome being the main indication.

On histology, a right-breast grade 3 invasive carcinoma was confirmed and comprised an NST carcinoma (Figure 2a,b) admixed with spindle cell components with squamous differentiation (Figure 2c,d). A diagnosis of metaplastic carcinoma was therefore made. There were areas of tumour necrosis and moderate lymphocytic infiltrate with associated lymphoid aggregates (Figure 2b). No lymphovascular invasion was seen, and the sentinel nodes were negative. The TNM stage was pT2 pN0sn pMx. The tumour was triple-negative and showed patchy strong positivity for CK14 and CK5/6, confirming a basal phenotype. MMR testing was undertaken, which showed that MSH6 and MSH2 were preserved with a deficiency of PMS2 and MSH1 (Figure 2e–h), a typical profile of Lynch syndrome.

The skin lesion on the contralateral breast appeared cystic macroscopically and was well defined in the deep dermis and subcutaneous tissue and covered by an intact epidermis.

Microscopically, the lesion was well circumscribed, seen in the dermis and extended to the subcutaneous tissue. The lesional cells comprised a proliferation of bland basaloid and clear cells and evidence of sebaceous differentiation, including mature sebocytes (Figure 3a–d). Occasional mitotic figures were noted, with no significant cytological atypia. Focal squamous differentiation was also seen. In places, vascular thrombosis with infarction and cholesterol crystals was noted. Immunohistochemistry showed positivity for AR, podoplanin, p63 and CK14 and patchy positivity for EMA. CK7 and CEA were negative. The appearances were those of a sebaceous neoplasm. Due to the large size and deep location of the lesion, a well-differentiated sebaceous carcinoma was considered. The diagnosis was discussed and agreed upon by skin pathology specialists. MMR molecular testing was undertaken, which, similar to the breast cancer, showed deficiency in MSH1 and PMS2, which was in keeping with the known Lynch syndrome.

## 3. Discussion

To our knowledge, this is the first report of a patient with co-existing breast cancer and sebaceous carcinoma of the breast with a background of Lynch syndrome. Table 1 summarises the current, though limited, literature on women with Lynch syndrome presenting with breast and/or skin manifestations.

There has been significant debate about the susceptibility to breast cancer in patients with Lynch syndrome. Currently, it is not listed in the cancers included in the Lynch syndrome spectrum. A cohort study by Therkildsen et al. showed that breast cancer was common in patients with Lynch syndrome [15]; this was supported by a case series from Brazil, which showed an earlier age of diagnosis of breast cancer in individuals with Lynch syndrome compared with the general population [16]. Conversely, a systematic review of molecular studies showed that only 51% of breast cancers had mismatch repair gene deficiencies in those with Lynch syndrome [17]. However, when investigating pathogenic variants in specific mismatch repair genes, *MSH6* and *PMS2* were associated with a statistically significant risk of breast cancer [18].

In the current patient, the breast cancer was triple-negative, which is similar to what has previously been reported for familial breast cancers, particularly in patients with germline pathogenic variants in the *BRCA1* germline [19]. The hormone receptor status of the breast carcinomas reported in patients with Lynch syndrome has been variable. A large cohort study showed that patients with mismatch-repair-gene-deficient invasive breast carcinomas were more likely to be ER- and PR-negative [20]. Conversely, in another series, 83% of Lynch syndrome-associated breast carcinomas were ER-positive; notwithstanding, 15 of the 18 cancers were HER2-negative [21].

Skin manifestations of Lynch syndrome have been reported in the literature. One variant of Lynch syndrome is Muir–Torre syndrome; this is defined as the presence of a sebaceous neoplasm alongside a Lynch syndrome-associated cancer. As expected, this is usually associated with colorectal cancer (in 50% of cases) [14]. Sebaceous neoplasms occurred following the detection of the visceral cancer in 59% of cases and prior to the detection of the visceral cancer in 41% of the cases [22]. In this current case, the patient had previously been diagnosed with colorectal cancer and, in this presentation, with breast cancer and a sebaceous carcinoma, which fits the definition of Muir–Torre syndrome. It is thought that this syndrome is rare and occurs in 9.2% of patients with Lynch syndrome [7].

The neoplasms reported in association with Lynch syndrome have included sebaceomas, sebaceous epitheliomas and carcinomas, basal cell carcinomas with sebaceous differentiation and cystic sebaceous tumours. Sebaceous tumours often present as papules or nodules, mostly on the face from the meibomian glands or in the extra-ocular region [23]. To our knowledge, this is the first case that reports sebaceous carcinoma of the skin of the breast with synchronous contralateral breast cancer associated with a prior Lynch syndrome diagnosis.

Recently, a Lynch syndrome transformation project has been established in England to ensure the standardisation of diagnostic and management pathways and minimise variation in care. This is likely to have a positive impact on the testing and treatment of patients with Lynch syndrome across various cancer types [24].

## 4. Conclusions

We present the first case of concomitant breast cancer and contralateral sebaceous carcinoma of the breast in a patient with a background of Lynch syndrome. This patient developed colorectal cancer, followed by metaplastic breast cancer and skin cancer. Awareness of the propensity of these patients to various types of cancers is essential for the managing multidisciplinary teams. Due to the increased risk of various cancers in these patients, regular screening should be undertaken, alongside molecular testing of any malignancy and cascade testing.

## Figures and Tables

**Figure 1 biomedicines-12-01242-f001:**
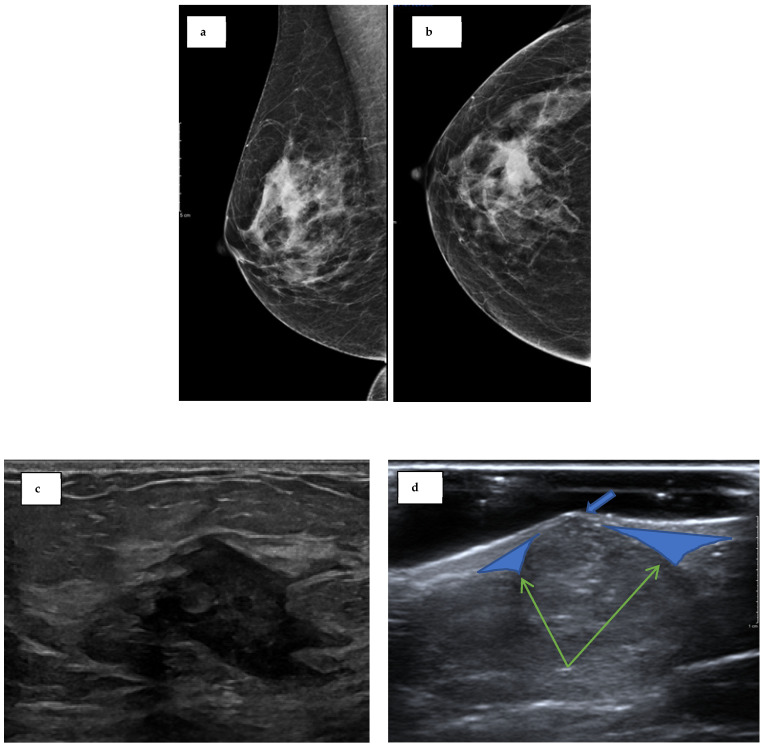
**Radiological presentation.** (**a**) Craniocaudal view of the right breast mammogram at symptomatic presentation. (**b**) Mediolateral oblique view of the right breast mammogram at symptomatic presentation. (**c**) Ultrasound of the right breast lump showing an irregular ill-defined malignant hypoechoic lesion (U5). (**d**) Ultrasound of the left breast mammographic finding. There is a punctum consistent with a sebaceous cyst (blue arrow). The claw sign (green arrows) finding indicates the lesion arose from the skin and not the breast.

**Figure 2 biomedicines-12-01242-f002:**
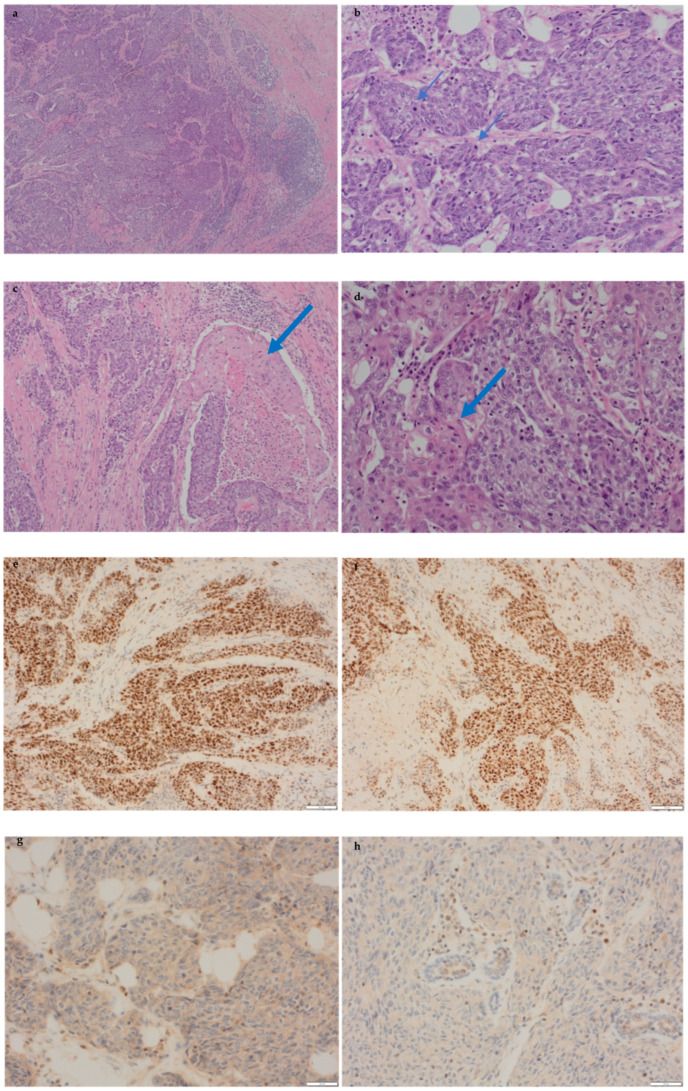
Histological appearances and immunohistochemistry for mismatch repair genes (MMRs) of the invasive breast carcinoma. (**a**) A well-circumscribed cellular tumour with adjacent lymphocytic infiltrate. (**b**) Confluent trabecula of malignant spindle-shaped cells with marked cytological atypia and conspicuous mitoses (thin blue arrows). (**c**,**d**) Poorly differentiated carcinoma with squamous differentiation (thick blue arrows). The tumour was MSH2- (**e**) and MLH6 (**f**)-positive (brown nuclei) and PMS2- (**g**) and MLH1 (**h**)-deficient (no brown nuclear staining).

**Figure 3 biomedicines-12-01242-f003:**
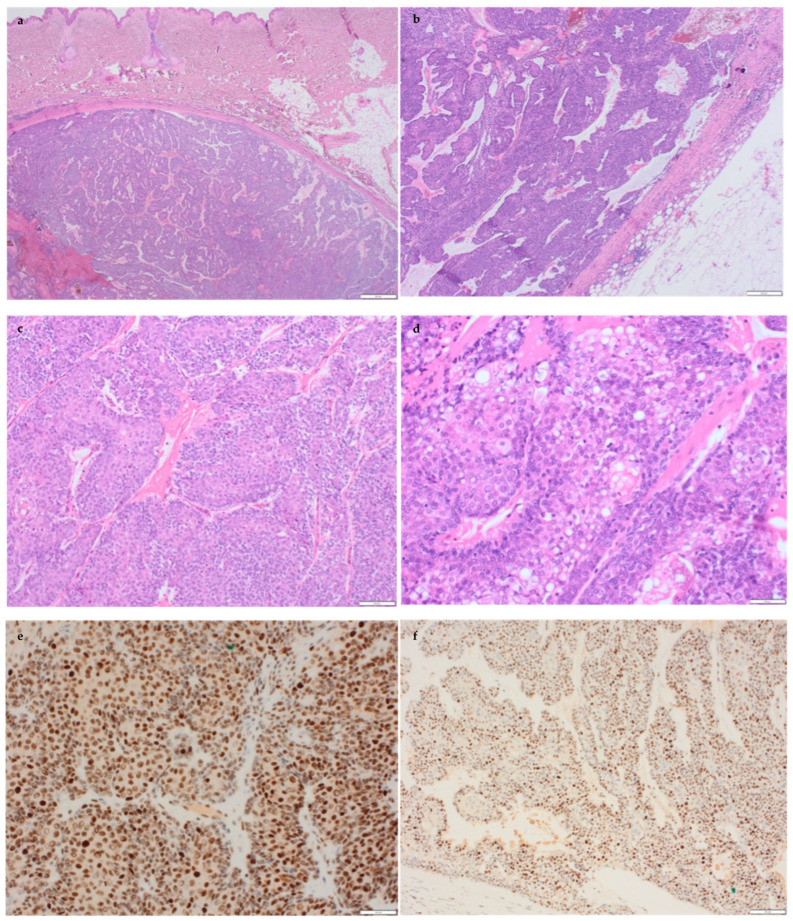
Histological features and MMR immunohistochemical profile of the left breast skin lesion. (**a**,**b**) Low-power view of the contralateral breast skin lesion showing a well-circumscribed deep dermal lesion with overlying normal skin. (**c**) A solid basaloid proliferation of mildly atypical cells. (**d**) The tumour showed evidence of sebaceous differentiation (clear cells). The tumour was MSH2- (**e**) and MLH6 (**f**)-positive (brown nuclei) and PMS2- (**g**) and MLH1 (**h**)-deficient (no brown nuclear staining).

**Table 1 biomedicines-12-01242-t001:** Reports of women with Lynch syndrome breast and/or skin manifestations.

Author	Year	Age	Previous Colorectal Cancer	Breast Neoplasm	Skin Neoplasm	Pathology Features	Genetic Features
Kientz et al. [11]	2017	40	No	High-grade in situ ductal carcinoma	Well-differentiated squamous cell carcinoma of the nose	Low expression of MSH2 and MSH6 in skin lesion	Germline deletion in *MSH2* gene
Kamisasanuki et al. [12]	2013	71	No	Historical	Sebaceous carcinoma of right eyelid, basal cell carcinoma of left eyelid	MSH2 gene absent in both tumours	-
Alzaraa et al. [13]	2008	43	No	-	Sebaceous carcinoma of the breast	-	Deletion in *MSH2* gene
Propeck et al. [14]	2000	46	Yes	-	Sebaceous carcinoma of the breast	Ill-defined lobules of basaloid cells with mitoses and focal calcifications	-

## Data Availability

The original contributions presented in this report are included in the article. Further enquiries can be directed to the corresponding author.

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
