# Peer review of "An Unusual Presentation of Synchronous Breast Cancer and Skin Malignancy in a Patient with Lynch Syndrome: A Case Report and Review of the Literature"

_biomedicines, 2024, doi:10.3390/biomedicines12061242_

Round 1

Reviewer 1 Report

Comments and Suggestions for Authors

Elghobashy et al. wrote a case report of breast cancer and skin cancer in a patient with Lynch syndrome and colorectal cancer history. The authors reported the diagnosis procedure and findings followed by detailed clinical presentation. The authors also reviewed the available reports of women with breast or skin cancer under the background of Lynch syndrome. Based on the report, it still seems to be controversial to list breast cancer as Lynch syndrome associated cancers. This is a nice report and will expand our knowledge about breast cancer/skin cancer and Lynch syndrome. The following issues can be resolved to improve the presentation of the manuscript.

Comments:

1.               Line 133: different fonts were used.

2.               The figures can be more informative. As a reader without much clinical background, it is hard to get the key information in Figures 2 and 3. More annotations like Figure 1 or more explanations in the figure legend may be helpful. For example, what do we expect to see if there are breast or skin malignancies in these figure panels?

Author Response

Thank you very much for your comments. We have amended the manuscript as per the comments below

  1. We have amended the fonts on line 133.
  2. Annotations have been added to Figure 2 and 3 with arrows and expanded figure legends (blue arrows and red text)

Reviewer 2 Report

Comments and Suggestions for Authors

The Authors present an interesting case report about breast cancer in a patient with Lynch syndrome. I recommend to accept the article after minor revisions.  

1. Can you please discuss more widely, if a mastectomy was necessary in this patient? Is the Lynch syndrome really the main indication for mastectomy or breast-conserving surgery can be performed? 

2. The authors state in conclusion section, that breast cancer should be added to the spectrum of Lynch syndrome malignancies. I think, that you can´t make this conlusion by a case report only. 

3. I am not sure, if you can state, that the breast cancer is a part of a Lynch syndrome, because it could be a sporadic cancer in patient with a Lynch syndrome and not due to a Lynch syndrome. Can you discuss this more widely? 

Author Response

Thank you very much for your comments.

  1. The main indication for mastectomy was Lynch syndrome.  With the confirmed Lynch syndrome by genetic testing, history of previous colonic cancer, current breast cancer and contralateral breast cancer risk, all surgical options were discussed with the patient. This included a discussion regarding breast conservation on the ipsilateral side followed by radiotherapy and some uncertainty about imaging follow up as well as a unilateral and bilateral mastectomy with or without an immediate reconstruction. The patient opted for bilateral mastectomies with no immediate reconstruction. This has been included in the text from line 85 (in red).
  2. and 3. 

    The immunohistochemical profile of the breast cancer in this patient confirmed mismatch repair gene deficiency typical of Lynch Syndrome. The profile was identical to her previous colorectal cancer and the concurrent skin cancer. Therefore, this is unlikely to represent a sporadic breast cancer (which should not show MMR deficiency).

    We agree that we cannot recommend inclusion bases on an individual case and we have tuned down the recommendation to “may be added” rather than “should be added” in the conclusion section to be similar to the abstract. We hope that this case report, will go some way to add to the evidence of breast cancer link to Lynch syndrome.